# Preferred versus Actual Place of Care and Factors Associated with Home Discharge among Korean Patients with Advanced Cancer: A Retrospective Cohort Study

**DOI:** 10.3390/healthcare11131939

**Published:** 2023-07-05

**Authors:** In Young Hwang, Yohan Han, Min Sun Kim, Kyae Hyung Kim, Belong Cho, Wonho Choi, Yejin Kim, Shin Hye Yoo, Sun Young Lee

**Affiliations:** 1Public Healthcare Center, Seoul National University Hospital, Seoul 03080, Republic of Korea; 83484@snuh.org (I.Y.H.);; 2Department of Family Medicine, Seoul National University Hospital, Seoul 03080, Republic of Korea; 3Department of Pediatrics, Seoul National University Hospital, Seoul 03080, Republic of Korea; 4Department of Medicine, Seoul National University College of Medicine, Seoul 03080, Republic of Korea; 5Center for Palliative Care and Clinical Ethics, Seoul National University Hospital, Seoul 03080, Republic of Korea

**Keywords:** neoplasms, palliative care, patient discharge, patient preference, retrospective studies, tertiary care centers

## Abstract

Respecting the preference for a place of care is essential for advance care planning in patients with advanced cancer. This retrospective study included adult patients with cancer referred to an inpatient palliative care consultation team at a tertiary acute care hospital in South Korea between April 2019 and December 2020. Patients’ preference for place of care and demographic and clinical factors were recorded, and the actual discharge locations were categorized as home or non-home. Patients discharged home but with unintended hospital visits within 2 months were also investigated. Of the 891 patients referred to the palliative care consultation team, 210 (23.6%) preferred to be discharged home. Among them, 113 (53.8%) were discharged home. No significant differences were found between patients who preferred home discharge and those who did not. Home discharge was higher among female patients (*p* = 0.04) and lower in those with poor oral intake (*p* < 0.001) or dyspnea (*p* = 0.02). Of the 113 patients discharged home, 37 (32.8%) had unintended hospital visits within 2 months. Approximately one-quarter of hospitalized patients with advanced cancer preferred to be discharged home, but only half of them received the home discharge. To meet patients’ preferences for end-of-life care, individual care planning considering relevant factors is necessary.

## 1. Introduction

Cancer is the leading cause of death in Korea [1]. However, advances in medical technology have increased the survival rate of patients with cancer [2]. While advances in medical technology have improved the cancer survival rates, patients with advanced cancer often receive aggressive treatment until the end-of-life (EOL), resulting in undue suffering, unrealistic expectations, and reduced opportunities to prepare for a dignified death [3]. Advance care planning (ACP) has been shown to promote hospice care and reduce futile healthcare utilization at EOL for patients with advanced cancer [4,5,6]. The Hospice and Palliative Care and Decisions on Life-Sustaining Treatment (LST) for Patients at the End of Life Act (abbreviated as “LST Decisions Act”), which was enacted in the Republic of Korea in February 2018, provides a legal basis for ACP that respects patients’ right to self-determination and promotes their best interests [7].

The knowledge of the preferred place of care is one of the most important components of ACP. Many patients prefer to be cared for or die at home, where their autonomy and privacy are respected, and home care consistent with the patient’s preference can reduce EOL medical costs [8]. A meta-analysis reported that 50–90% of patients with cancer wished to be cared for or die at home [9]. However, 87.2% of patients with terminal cancer spent time in a hospital at the EOL [10]. To respect the patient’s preference, it is important to know the barriers against achieving care at home at the EOL. Previous studies have suggested that the patient’s will, the presence of physicians who are aware of the patient’s preference, and caregivers’ support are positively associated with home deaths [11,12,13,14]. Current evidence predominantly focuses on the preferred place of death rather than the preferred place of care [14,15]. Furthermore, there is scarce literature regarding the number of patients with cancer who wish to be discharged home after a diagnosis of advanced cancer with limited prognosis while being in hospital and the factors associated with home discharge.

The primary aim of this study was to determine the preferred and actual places of care for patients with advanced cancer and identify the factors associated with their home discharge from a tertiary hospital.

## 2. Materials and Methods

### 2.1. Study Design and Patients

This retrospective study analyzed the data of hospitalized patients with advanced cancer at the Seoul National University Hospital (SNUH). The SNUH is a 1761-bed tertiary referral hospital in the Republic of Korea, staffed by 1947 physicians working in acute and specialized care. Among the hospitalized patients, those who were referred to an inpatient palliative care consultation team at the SNUH between April 2019 and December 2020 were included. Patients who had already passed away before meeting with the team, declined consultation with the team, or had insufficient information regarding their preferred place of care in the chart review were excluded. 

In the Republic of Korea, three types of hospice palliative care services have been implemented in the healthcare system, which is supported by the national health insurance: (1) inpatient hospice care (ward), (2) home hospice, and (3) inpatient palliative care consultation [16]. No other community-based palliative care service exists in the healthcare system. The palliative care service at the SNUH, which includes an outpatient clinic and an inpatient palliative care consultation team, does not provide inpatient hospice-palliative care ward services or home hospice-palliative care services. No standardized criteria have been used for the referral for consultation to date. The inpatient palliative care consultation team at the SNUH consists of a palliative care physician who specializes in hematology and medical oncology, palliative care nurses, and medical social workers. The team conducts a comprehensive assessment of the patients’ and caregivers’ physical, psychosocial, and spiritual needs, including the preferred place of care. They also explore the individual’s own preferences and values and conduct goals-of-care discussion and advance care planning for EOL through thorough interview with the patient and caregivers. Then, the team delivers the interview contents and recommendations for palliative care planning to the primary attending physician who makes a referral, facilitating collaborative decision making among all the stakeholders [17,18].

### 2.2. Data Collection and Measurement

We reviewed electronic medical records and used data retrieved from the SNUH Patient Research Environment system. We collected sociodemographic data, such as age, sex, religion, marital status, insurance, residence, caregiver’s relationship with the patient, and the number of family members living together. The healthcare reimbursement system in the Republic of Korea has two components: (1) the national health insurance, which provides coverage to all citizens, is managed comprehensively in the form of social insurance and is funded by beneficiaries’ contributions and (2) medical aid that provides support to lower income groups, funded by the general revenue. We also collected clinical data, including the cancer type, treatment status, and medical conditions. Hepatobiliary and pancreatic cancers were separately classified from other gastrointestinal cancers because they often require therapeutic interventions, such as endoscopic or percutaneous drainage procedures, which may result in differences in hospitalization and place of care compared to other gastrointestinal cancers. Medical conditions included the presence of symptoms, such as state of consciousness, dyspnea, pain, nausea, vomiting, poor oral intake, urinary symptoms, fever, or bleeding, and included medical needs, such as oxygen demand, dependency on total parenteral nutrition, or opioid demand using a morphine equivalent dose, assessed at the time of referral and discharge. We also assessed the presence of multidrug-resistant (MDR) bacterial organisms at discharge, including methicillin-resistant *Staphylococcus aureus* as well as vancomycin-resistant *Enterococcus*, carbapenem-resistant *Enterobacteriaceae*, and carbapenem-resistant *Pseudomonas aeruginosa*. The preferred place of care was assessed at referral, and the discharge outcome (home, hospital, or death) was also collected.

As the LST Decisions Act was recently enacted, the completion of related documents after a series of sufficient discussion has become one of the important steps during ACP. We investigated the presence and type of documents, such as the Advance Directives (AD) or Physicians’ Orders for LST (POLST), the legal documentation for LST implementation according to the LST Decisions Act, and the dates of documentation. Patients with decision-making capacity were permitted to sign an AD or POLST. According to the LST Decisions Act, to implement the decisions to withhold or withdraw LST, the attending physician and another relevant specialist jointly evaluate the patient’s medical condition and assess whether the patient was at the “EOL process”. The “EOL process”, a term mentioned in the LST Decisions Act, is a legal prerequisite for implementation of LST decisions, defined as a state of imminent death despite treatment. The Korean consensus guideline [19] proposed the use of “the last days of life” to aid clinical judgment for the “EOL process”. If the patient’s intention not to perform LST could not be identified, the family was allowed to make surrogate decisions by assuming the patient’s intention or unanimous agreement on LST. LST implementation according to the AD or POLST was defined as self-determination.

For patients discharged home, we investigated healthcare use within 2 months after discharge, including visits to the emergency department (ED) of the SNUH and unscheduled hospitalization, defined as admission to the SNUH, not for scheduled chemotherapy or planned procedures. Unintended hospital visits within 2 months after discharge were defined as either any ED visit to the SNUH or unscheduled hospitalization to the SNUH within 2 months after the discharge date. The study utilized data that had been collected up to 31 July 2021.

### 2.3. Statistical Analysis

To illustrate the baseline characteristics of the patients in the study, descriptive analyses were conducted using median and interquartile ranges (IQR, Q1 to Q3) for the values and percentages to represent the number of patients.

We classified the preferred and actual places of care into two groups: home and non-home discharge (hospital discharge or death). We used different statistical tests depending on the type of data: Student’s *t*-test or Mann–Whitney U test for continuous variables (parametric and non-parametric analyses, respectively) and Chi-square test or Fisher’s exact test for categorical variables to compare the data between the two groups. We analyzed the status of the advanced statement for patients who preferred home as their care place, while the legal documentation status for LST implementation was only analyzed for deceased patients. All statistical tests were two-sided, and *p*-values < 0.05 were considered statistically significant. We did not perform any imputations for the missing data, as none of the variables had missing values exceeding 5%. The analyses were conducted using the R package (moonBook) of R version 3.6.0 (R Core Team (2019), R: Language and environment for statistical computing).

## 3. Results

Out of the 1071 patients who were enrolled during the study period, we excluded 30 patients who were not assessed by the research team and 150 patients who either had missing information or incomplete data on their preferred place of care (Figure 1). A total of 891 patients were finally included.

### 3.1. Patient Characteristics According to a Preferred Place of Care

Table 1 presents the demographic and clinical characteristics of 891 patients with advanced cancer. The median age was 64.5 years, and 55.3% of patients were male. Among patients, 23.6% (*n* = 210) preferred home as their place of care. Only a small proportion of patients (3.4%) had medical aid insurance. The most common single cancer type was hepatobiliary cancer, followed by lung and gastrointestinal cancers. No significant differences were observed in terms of demographic and clinical characteristics between patients who preferred home as a place of care and those who preferred hospital. The median hospitalization duration was 18 (IQR, 10–30) days for all patients, 15 (IQR, 9–28) days for those who preferred home care, and 20 (IQR, 10–30) days for those who preferred hospital care.

### 3.2. Discharge Location among Patients Who Preferred Home as a Place of Care and the Factors Associated with Home Discharge

Among the 210 patients who preferred home as a place of care, 53.8% (*n* = 113) were discharged, 22.9% (*n* = 48) were transferred to another hospital, and 23.3% (*n* = 49) passed away at the SNUH. 

Table 2 and Appendix A show the association between sociodemographic and clinical factors and actual home discharge among the 210 patients. Female patients were significantly more likely to be discharged home compared to those who were not discharged home (48.7% vs. 34%, respectively; *p* = 0.04). Only a small proportion of patients (2.1%) lived alone, and no patients who preferred home lived alone. Spouses were the main caregiver (62.3% vs. 67.6% vs. 56.8%, *p* = 0.15); no significant differences were found in terms of caregiver characteristics, such as spouse as the main caregiver, age, residence, insurance, marital status, or religion, between patients who preferred home, those who were discharged home, and those who were not.

More patients who were still on chemotherapy were discharged home than off-chemo patients (48.7% vs. 22.9%, *p* < 0.001). Outpatient admissions were significantly associated with home discharge compared to ED admissions. (61% vs. 54%, *p* = 0.04). The proportion of cancer types was similar in both groups. The top three common major medical problems at the time of referral in the overall patient group were pain (73.2%), poor oral intake (53.3%), and dyspnea (35.2%). Poor oral intake was negatively associated with home discharge (*p* < 0.001), and patients with dyspnea were less likely to be discharged home than those without (*p* = 0.02). MDR pathogen prevalence was lower among patients discharged home, but the difference was not statistically significant (2.7% vs. 9.3%, *p* = 0.70). The total number (%) of patients with any opioid use was 58 (27.6%); among them, 37 (32.7%) patients were discharged home and 21 (21.6%) were not. Among the opioid users, the median opioid doses (morphine equivalent dose, mg) in the home and non-home discharge groups were 65.5 (IQR, 32.5–133) and 75 (IQR, 36.5–203) mg, respectively (*p* = 0.41).

### 3.3. Status of Documentation Related to the LST Decisions Act among Patients Who Preferred Home as the Place of Care

Figure 2 illustrates the documentation status of advance statements, such as AD or POLST, who preferred a home discharge. Among the 210 patients, 62.1% (*n* = 143) had documented AD (*n* = 16) or POLST (*n* = 127). The proportions of patients with AD or POLST in the home discharge, the hospital discharge, and the hospital death groups were 71.7% (*n* = 81), 60.4% (*n* = 29), and 67.3% (*n* = 33), respectively, with no statistically significant differences between the groups *(p =* 0.37). One half (52%, *n* = 35) of the 67 patients who were discharged home documented an advance statement after discharge.

Figure 3 presents the status of the legal documentation form for LST implementation on deceased patients who preferred a home discharge. The number of deceased patients was 187 out of 210 patients (89.0%): 95 out of 113 in the home discharge group (84.0%) and 43 out of 48 in the hospital discharge group (89.6%).

The median follow-up duration was 44 (IQR, 21–88.5) days in the deceased patients: 85 (IQR, 24–187.6) days in the deceased home discharge group, 53 (IQR, 35–71) days in the deceased hospital discharge group, and 23 (IQR, 13–38.5) days in the hospital death group. The rate of documenting LST was significantly different between the home discharge (36.9%), the hospital discharge (60.5%), and the hospital death groups (93.9%) (*p* < 0.001).

### 3.4. Post-Discharge Outcomes of Patients Who Preferred and Went Home

Of the 113 patients who preferred home as a place of care and were discharged, 26.5% visited the ED, 8% had an unscheduled hospitalization, and 32.7% had unintended hospital visits within 2 months following discharge (Table 3). The median times from discharge to the event were 16 (IQR, 9.3–28.5) days, 19 (IQR, 11–25) days, and 16 (IQR, 9–25) days for the ED visits, unscheduled hospitalization, and unintended hospital visits, respectively. The median follow-up duration from the referral date of the inpatient palliative care consultation of the 113 patients was 81 (IQR, 38–149) days. No factors, except for the status of continuing chemotherapy, were associated with unintended hospital visits (Appendix A).

## 4. Discussion

In this study, we examined the preferred and actual places of care for patients with advanced cancer and reported the factors associated with their home discharge from a tertiary hospital. EOL care for patients with cancer usually takes place in acute care hospitals specializing in cancer treatment. Inpatient palliative care consultation is important in discharge planning and considering patients’ preferred place of care [20]. In our study, only a quarter of patients met the criteria for home care preference, and only half of them were actually discharged home. Female patients were more likely to be discharged home, while poor oral intake and dyspnea were negatively associated with home discharge. The comparison of the results of this study to those of previous studies is limited because previous studies focused on the preferred place of death, whereas this study investigated the preferred place of care following discharge. Our findings are consistent with those of a similar study conducted in the United Kingdom [21], which revealed that nearly half of the patients who expressed a preferred place of care did not die in their desired location, and that the place of care preferences may change over time. However, the proportion of patients who preferred home discharge was slightly lower compared to the corresponding proportion in a previous Korean study [13]. This could be explained by the fact that hospitalized patients with advanced cancer, especially in acute-care settings, typically have more severe symptoms and may feel more anxious [22]. These findings provide an insight into the preference for a place of care for EOL and the possible barriers to a home discharge. Inpatient palliative care consultation services can improve post-discharge outcomes with care coordination before discharge [20] but cannot follow-up patients at home [23]. Therefore, it can be beneficial to link patients who want a home discharge to adequate services providing continuous medical care, such as home-based palliative care.

Previous studies have reported that there was no correlation between sex [24,25,26] and the preferred place of care. However, another study reported that female patients with cancer were more likely to be discharged home than male patients [10]. In line with these findings, our study found that female patients were more likely to be discharged home. This can be explained by the fact that Korean women (especially older women) may feel more familiar with living at home for a long time as homemakers and prefer staying at home [27]. However, a recent report indicated that the actual home discharge rate was lower among older women [28]. The influence of sex on congruence between preferred and actual home care warrants further investigation.

This study revealed that dyspnea and poor oral intake were associated with non-home discharge among patients who preferred home as the place of care. Dyspnea is a challenging symptom that cannot be resolved completely despite an opioid prescription at home and may cause high levels of anxiety [29], resulting in patients dying in hospitals [25]. Persistent poor oral intake, a common symptom in patients with advanced cancer, usually requires parenteral support, including hydration at home, to prevent accompanying problems, such as dehydration, electrolyte imbalances, and physical deterioration [30]. Proper symptom management at home, including home visits for parenteral approaches, may help patients with poor oral intake stay at home longer [15].

The positive association of being on chemotherapy with home discharge suggests that patients who receive chemotherapy are more likely to be referred at an earlier point in the disease trajectory. Thus, they would be more likely to stay at home for a longer period in a better functional status and to use fewer subsequent healthcare resources. Owing to the limitations of the retrospective study design, the functional status was not investigated. However, it is believed that the chemotherapy status can indirectly reflect this.

One-third of patients discharged home had unplanned hospital visits within 2 months, with insufficient symptom management and a lack of support being major concerns [31]. Home-based medical care has been shown to improve post-discharge outcomes [32], but Korea lacks transitional palliative care services for these patients. Considering the fact that the median time from discharge to an unintended hospital visit was 16 days in this study, transitional care during the immediate post-discharge period for these patients may help prevent unnecessary visits.

This study is the first in Korea to report on the status of documentation of advance statement and LST implementation for patients with cancer who prefer to receive care at home after the LST Decisions Act was enforced. The results showed that 68% of these patients had documented advanced statements, such as AD or POLST, which is consistent with the national data. It was observed that some patients wrote the document after discharge, which may be attributed to the hospital environment’s not being conducive to discussing ACP after hospitalization for acute illness [33]. However, some patients may not take ACP seriously or consider it a loss of hope after discharge. Therefore, it is crucial to facilitate the communication concerning ACP by clarifying personal goals, values, or preferences and to help patients prepare documentation. Moreover, our study found that only 57.2% of patients who preferred home discharge and eventually died had implemented LST decisions through written documents, which is lower than the rate reported in another study of hospitalized patients at the same institution [6]. For patients discharged home, this rate decreased by 36.9%. It is possible that patients may have died in settings that lack the ability to implement LST decisions [34]. Further research is needed to investigate whether a patient’s wishes are respected and implemented after a home discharge.

This study has several clinical implications. First, the current status of place of care and of a discharge at the first transitional phase to hospice palliative care, which is demonstrated in this study, provides valuable basic data for understanding the correlations of the staying-home congruent with a patient’s desire based on the care trajectory of patients with advanced cancer. Given that home-based medical care is limited to a few home hospices in Korea, this study suggests that home-based medical services tailored to patients’ needs should be expanded to prevent unnecessary hospital visits after home discharge. Furthermore, as the related documentation to the LST Decisions Act has been investigated in hospital-based settings in Korea, our findings may facilitate the process of application of the LST Decision Act in patients receiving home-based palliative care.

However, the study has some limitations. It was conducted at a single center; therefore, the results may not be generalizable to all patients with advanced cancer. As it was a retrospective study, we could not capture some variables, which may have influenced the home discharge congruent with the preference on home discharge, such as systematically evaluated symptom profiles, functional status, changes in patient preferences during hospitalization, or the family caregiver’s preferences. Furthermore, the study only examined healthcare utilization at one institution, which may not represent overall healthcare utilization. Future prospective studies are needed to track the patient trajectory from discharge to death.

## 5. Conclusions

In conclusion, this study highlights that a considerable proportion of hospitalized patients with advanced cancer expressed a preference for home discharge, but only a fraction of them actually received it. Sex was found to be a significant factor in actual home discharge, while symptoms, such as poor oral intake or dyspnea, were identified as barriers. To better meet the needs and fulfill the desires of these patients for a specific place of care, further research and development of home-based medical services tailored to their needs may be necessary.

## Figures and Tables

**Figure 1 healthcare-11-01939-f001:**
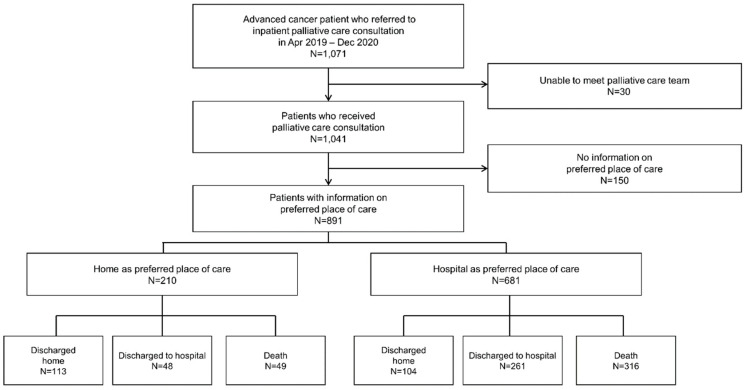
Flow chart of study participants.

**Figure 2 healthcare-11-01939-f002:**
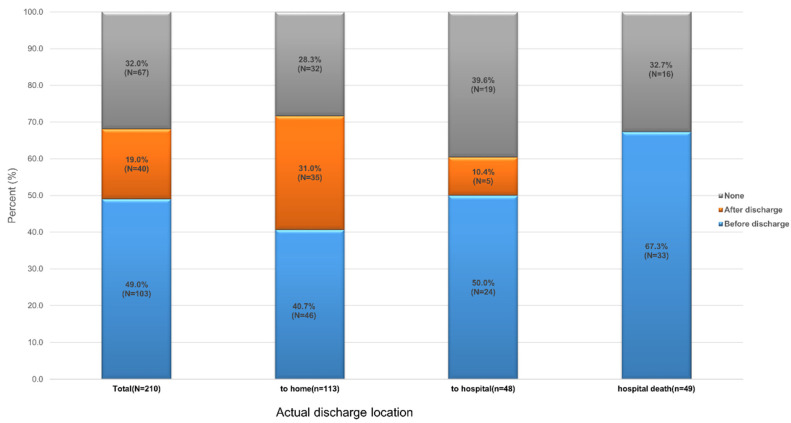
Status of advance statement documentation and timing of documentation according to the actual discharge location for 210 patients who preferred home discharge.

**Figure 3 healthcare-11-01939-f003:**
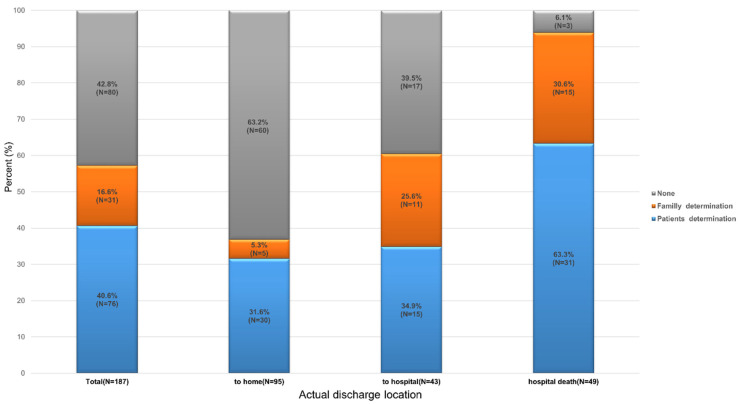
Status of legal documentation for life-sustaining treatment implementation according to actual discharge location for 187 deceased patients who preferred home discharge and died during or after hospitalization.

**Table 1 healthcare-11-01939-t001:** Baseline demographic and clinical characteristics of hospitalized patients with cancer with preferred place of care (*n* = 891).

Variables	Total (*n* = 891)		Patients Who Preferred Home as Place of Care (*n* = 210)		Patients Who Preferred Hospital as Place of Care (*n* = 681)		*p*-Value
	*n*	%	*n*	%	*n*	%	
**Age**							0.73
Median, years (IQR)	64.5 (57.2–73)		64 (56.2–73)		64.6 (57.3–72.9)		0.77 *^a^*
<45	75	8.4	20	9.5	55	8.1	
≥45, <65	385	43.2	87	41.4	298	43.8	
≥65	431	48.4	103	49.0	328	48.2	
**Sex**							0.40
Male	493	55.3	122	58.1	371	54.5	
Female	398	44.7	88	41.9	310	45.5	
**Residence**							1.00
Metropolitan	561	63	132	62.9	429	63.0	
Non-metropolitan	330	37	78	37.1	252	37.0	
**Medical insurance**							0.19
National health insurance	862	96.7	203	96.7	659	96.8	
Medical aid	29	3.3	7	3.4	22	3.2	
**Type of cancer**							0.60
Lung	153	17.2	42	20	111	16.3	
Breast	53	5.9	11	5.2	42	6.2	
Gastrointestinal	138	15.5	36	17.2	102	15	
Hepatobiliary-pancreas	221	24.8	53	25.2	168	24.7	
Hematologic malignancy	85	9.5	16	7.6	69	10.1	
Others *^b^*	241	27	52	24.8	189	27.7	
**Duration of admission (days)**							
Median (IQR)	18 (10–30)		15 (9–28)		20 (10–30)		0.04 *^a^*
**Time from assessment to discharge (days)**							
Median (IQR)	7 (3–14)		7 (3–14)		10 (3–13)		0.75 *^a^*

Abbreviation: IQR, interquartile range. *^a^* Mann–Whitney U test. *^b^* Others include malignant neoplasms of the lip, oral cavity and pharynx, thyroid and other endocrine glands, eye, brain, other parts of the central nervous system, skin, mesothelial and soft tissue, male and female genital organs, urinary tract, and sites that are ill-defined or unspecified and other secondary sites.

**Table 2 healthcare-11-01939-t002:** Characteristics of patients who preferred home discharge according to actual discharge location.

Variables	Total (*n* = 210)		Discharged Home (*n* = 113)		Not Discharged Home (*n* = 97)		*p*-Value
	*n*	%	*n*	%	*n*	%	
**Age, years**							0.04
Median (IQR)	64 (56.2–73)		64 (53–73)		67 (59–73)		0.16 *^a^*
<45	20	9.5	13	11.5	7	7.2	
≥45, <65	87	41.4	49	43.4	38	39.2	
≥65	103	49	51	45.1	52	53.6	
**Sex**							0.04
Male	122	58.1	58	51.3	64	66.0	
Female	88	41.9	55	48.7	33	34.0	
**Residence**							0.88
Metropolitan	132	62.9	70	61.9	62	63.9	
Non-metropolitan	78	37.1	43	38.1	35	36.1	
**Medical insurance**							0.26
National health insurance	201	95.7	106	93.8	95	97.9	
Medical aid	9	4.3	7	6.2	2	2.1	
**Marital status**							0.80
Married	164	72.9	87	77.0	77	79.4	
Divorced/bereaved/single	43	21	26	23.0	20	20.6	
**Religion** *^b^*							0.29
Yes	146	74.1	71	67.6	75	81.5	
**Number of co-habiting family members** *^c^*							0.11
0	4	2.1	0	0.0	4	4.3	
1	94	49.2	49	49.5	45	48.9	
≥2	93	48.7	50	50.5	43	46.7	
**Relationship of main caregiver** *^d^*							0.15
Spouse	129	62.3	75	67.6	54	56.8	
Non-spouse	77	37.4	36	32.4	41	43.2	
**Type of cancer** *^e^*							0.64 *^f^*
Lung	42	20	23	20.4	19	19.6	
Breast	11	5.2	7	6.2	4	4.1	
Gastrointestinal	36	17.1	20	17.7	16	16.5	
Hepatobiliary-pancreas	53	25.2	24	21.2	29	29.9	
Hematologic malignancy	16	7.6	11	9.7	5	5.2	
Other	52	25.2	28	24.8	24	24	
**Treatment status**							<0.001
On chemotherapy	77	37.4	55	48.7	22	22.9	
Off chemotherapy	132	62.9	58	51.3	75	77.1	
**Route of admission**							0.04
Via outpatient clinic	82	39	52	46.0	30	30.9	
Via emergency department	128	61	61	54.0	67	69.1	
**Major medical problem**							
Confusion	19	9	7	6.2	12	12.4	0.15 *^f^*
Pain	153	73.2	80	71.4	73	75	0.64
Fever	35	16.7	16	14.2	19	20	0.37
Dyspnea	74	35.2	31	27.4	43	44	0.02
Nausea/vomiting	36	17.1	18	15.9	18	19	0.75
Poor oral intake	112	53.3	47	41.6	65	67	<0.001
Urologic symptoms	10	4.8	6	5.4	4	4.1	0.75 *^f^*
Bleeding (any)	11	52	5	4.4	6	6.2	0.76 *^f^*
MDR pathogens	12	5.7	3	2.7	9	9.3	0.70 *^f^*

Abbreviations: MDR, multidrug resistant. *^a^* Mann–Whitney U test. *^b^* Number of missing data = 13. *^c^* Number of missing data = 19. *^d^* Number of missing data = 4. *^e^* Others include malignant neoplasms of the lip, oral cavity and pharynx, thyroid and other endocrine glands, eye, brain, other parts of the central nervous system, skin, mesothelial and soft tissue, male and female genital organs, urinary tract, and sites that are ill-defined or unspecified and other secondary sites. *^f^* Fisher’s exact test.

**Table 3 healthcare-11-01939-t003:** Post-discharge hospital visits of patients who preferred and were discharged home.

Variables	Total (*n* = 113)	
**ED visit after discharge within 2 months**		
*n* (%)	30	26.5
Time from discharge to ER visit (days), median (IQR)	16	9.3–28.5
**Unscheduled readmission (to tertiary hospital) after discharge within 2 months**		
*n* (%)	9	8.0
Time from discharge to readmission (days), median (IQR)	19	11–25
**Unintended hospital visit within 2 months** *^a^*		
*n* (%)	37	32.7
Time from discharge to unscheduled hospital visit (days), median (IQR)	16	9–25

Abbreviations: ED, emergency department; IQR, interquartile range; SNUH, Seoul National University Hospital. *^a^* Unintended hospital visit within 2 months was defined as either any visit to the ED of SNUH or unscheduled hospitalization (not for scheduled chemotherapy or procedure) to SNUH within 2 months after the discharge date.

## Data Availability

The data presented in this study are available in the manuscript.

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
