# Peer review of "Preferred versus Actual Place of Care and Factors Associated with Home Discharge among Korean Patients with Advanced Cancer: A Retrospective Cohort Study"

_healthcare, 2023, doi:10.3390/healthcare11131939_

Round 1

Reviewer 1 Report

The manuscript contains information that justifies publication. The title, although a little long, is consistent with the purpose of the study and the methods employed. The summary presents enough information to understand the procedures performed and what was performed. Considering that the descriptors are not enough for the identification and visibility of the manuscript, I suggest inserting another one or replacing one of them. The introduction is clear, presents a logical sequence, and justifies the importance of the study. The Method is consistent with the title and objective of the study. The description of the methodological procedures is adequate for the type of study. The place and period of the study and the participants involved were adequately described (the authors could have better explained the inclusion and exclusion criteria). The technique for data collection was adequately described and the analyzes carried out are adequate. The authors could have commented on the ethical aspect involved, for example, the waiver of the consent form, since the study was based on a secondary source. In the description of the results, the Flow Chart was very enlightening. The results are limited to the findings obtained and the tables presented facilitate the understanding of the data. The results are limited to the findings obtained and the tables presented facilitate the understanding of the data. The discussion addresses the relationship between the data obtained in this study and the national literature, but the authors could have presented a better relationship between their results and the international literature. The authors present the limitations of the study but could have better emphasized the contributions of their study. The conclusion responds to the proposed objective and is based on the results and is consistent with the title and method. References are adequate and 50% of them date from the last five years.

Reviewer 2 Report

Thanks to the authors for their efforts to elucidate factors associated with discharge home in hospitalized cancer patients who wish to return home.

In order to assist the reader with interpreting the results, the following revisions are suggested:

1. I would strongly recommend replacing "terminal" cancer with "advanced" cancer or "incurable" cancer throughout the paper.  "Terminal" is an imprecise term which may be interpreted as "untreatable", which is not the case with this study population (some patients were receiving chemotherapy).

2. Please provide more context regarding palliative care services and other aspects of the healthcare system.  What are the referral criteria for the inpatient palliative care consultation team?  Who is on the team?  Who can make a referral?  It would be helpful to mention in the Methods that community-based palliative care services are limited, rather than only in the Discussion.  What is Medicaid?  What may be reasons for discharging to another hospital?

3. How were symptoms determined retrospectively?  Was a tool used to systematically document symptoms during admission?  Was functional status assessed?

4. Hepatobiliary-pancreas cancers are included in gastrointestinal cancers.  Please clarify if there is a specific reason to list hepatobiliary cancers separately.  

5.  Morphine equivalent dose is listed in the methods but not presented in the results - please add.

6.  In Section 3.3., does "median follow-up duration...in the deceased patients" refer to survival from time of inpatient palliative care consultation?

7.  One factor in preference for and actual home discharge is not explicitly mentioned, although implied in the results; that is, patients preferring and actually discharged home were likely at an earlier point in the disease trajectory.  Thus, they had a lower rate of death in hospital, had shorter duration of admission, were more frequently on chemotherapy and admitted from an outpatient clinic, and less frequently experiencing poor oral intake and dyspnea.  Please discuss.

8. Persistent poor oral intake is expected as the patient approaches the end of life.  In this case, parenteral nutrition would not be beneficial - please clarify in the Discussion.

9. The sentence on lines 57 to 59 includes a repetition and is not a complete sentence.

10. The title of Figure 3 could be clearer, for example, "Status of legal documentation for life-sustaining treatment implementation according to actual discharge for 187 patients who preferred home discharge and died during or after hospitalization."

Reviewer 3 Report

Abstract

Of the 891 patients referred to the palliative care 24 consultation team, 210 (23.6%) preferred to be discharged home, but only 113 (53.8%) were actually discharged home --- the percentage referred to 113 should be on the total, thereby 12.7% not 53.8%

Several typos and grammatical errors across the manuscript – e.g., line 43 - missed space between words, incorrect modality of citation, comma between subject and verb.

Introduction

Line 49. It is the KNOWLEDGE of the preferred place of care one of the pivotal components of ACP, not just the preference on the place of care. When the place of care is known, it is supposed that discussion around end of life had took place.

Methods.

Line 75-77. It seems to me that exclusion criteria do not fit with the study design. This is a retrospective study, thus, how can patients refuse participation? How did authors ascertain if patients have enough information?

Line 94-95. “To identify ACP status, we investigated the presence of documentation of advanced statements”. This is conceptually wrong: ACP cannot be limited to the documentation of ADs, it is a complex process that also includes preparing people and surrogate decision maker for communication and decision-making. There is a lot of literature around ACP vs AD (see J palliative Medicine 2022, Caught in a loop with advance care planning and advance directive). Anyway, I do not understand this insistence and continuing referring to ACP. This paper is not focused on ACP.

Line 98-99. How doctors assessed whether the patient was in the EOL process? What do author mean with EOL process? The last days, weeks, or months?

Results

Which is the meaning to split patients’ information into table 2 and 3. This should appear altogether.

Section 3.3. In my opinion, this section is conceptually incorrect as per my previous comment on the definition authors employed for ACP. The data presented in this section cannot be defined ACP.

Discussion

Line 239- 248 provide conflicting interpretation of the study findings on the relationship between gender and preferred/actual place of care.

Line 265-280. There is confusion between ACP and AD that are used interchangeably as they were synonym but there are not.  

English is not fluent, there are several repetitions and the meaning of some sentences is difficult to grasp, particularly in the introduction and the discussion. The paper would benefit from review by a native English-speaking editor.

Round 2

Reviewer 2 Report

Thank you for the revisions, which have made the manuscript easier to understand.  My only remaining suggestion is to explain "medical aid" in the text, as it would be unclear to many readers that it is funding for patients in lower income groups.

Author Response

#1. Thank you for the revisions, which have made the manuscript easier to understand.  My only remaining suggestion is to explain "medical aid" in the text, as it would be unclear to many readers that it is funding for patients in lower income groups.

  • Thank you for your comment. As suggested by the reviewer, we added the explanation for ‘medical aid’ in the manuscript as follows (page 4).

Healthcare reimbursement system in the Republic of Korea has two components: 1) national health insurance to provide coverage to all citizens, managed comprehensively in the form of social insurance and funded by beneficiaries’ contributions; 2) medical aid to provide support to lower income groups, funded by general revenue.

Reviewer 3 Report

I'd thank the authors for their efforts in addressing each of the reviewers' comments. I have no further suggestions, except for having English revised.

As declared by the authors, the English editing is planned after this revision process.

Author Response

Thank you for your efforts to review our manuscript. We have done English proofreading again after the first revision process.